# Pilot Study: The Effectiveness of Hyperbaric Oxygen Therapy in the Treatment of Periodontitis in Patients with Type 2 Diabetes

**DOI:** 10.3390/healthcare11091344

**Published:** 2023-05-07

**Authors:** Katarzyna Latusek, Adrianna Słotwińska-Pawlaczyk, Aleksandra Warakomska, Magdalena Kubicka-Musiał, Rafał Wiench, Bogusława Orzechowska-Wylęgała

**Affiliations:** 1Department of Pediatric Otolaryngology, Head and Neck Surgery, Chairs of Pediatric Surgery, The Independent Public Clinical Hospital No. 6 of the Medical University of Silesia in Katowice, John Paul II Upper Silesian Child Health Centre, 40-055 Katowice, Poland; 2Department of Periodontal Diseases and Oral Mucosa Diseases, School of Medicine with the Division of Dentistry in Zabrze, Medical University of Silesia in Katowice, Pl. Traugutta 2, 41-800 Zabrze, Poland

**Keywords:** hyperbaric oxygen therapy, diabetes, periodontitis, periodontal disease

## Abstract

Periodontitis is a chronic inflammatory disease with multifactorial aetiology. The relationship between periodontal disease and systemic diseases such as diabetes, obesity, metabolic syndrome, atherosclerotic, cardiovascular disease, and cognitive disorders has been the subject of many studies. The purpose of this study was to evaluate the effectiveness of hyperbaric oxygen therapy on periodontal health in patients suffering from periodontitis and type 2 diabetes. The study was conducted with 14 patients. A total of 369 periodontal pockets in the study group and 431 in the control group were examined. For further analysis, the pockets were classified as moderately deep (4–5 mm) and deep (≥6 mm). All patients received standard non-surgical treatment: scaling and root planing (SRP). Additionally, a series of 30 hyperbaric chamber sessions was carried out in the study group. The following parameters were compared between groups: PD (probing depth), CAL (clinical attachment level), and BOP (bleeding on probing). The results of the study showed significantly better results in terms of PD reduction and CAL gain in the study group in comparison to the control group. Both groups showed a reduction in BOP (bleeding on probing) after treatment. The use of hyperbaric oxygen therapy seems to have considerable benefits in patients with type 2 diabetes.

## 1. Introduction

According to the European Oral Health Report, more than 50% of the European population may suffer from some form of periodontitis. Additionally, more than 10% suffer from a severe form of the disease, with the prevalence increasing to 70–85% of the population aged 60–65 years [1]. The Global Burden of Disease Study (2016) ranked periodontal disease as the 11th most prevalent condition worldwide [2]. Alawaji et al. (2022) assessed the frequency of periodontal disease in 431 individuals with a mean age of 35.4 (13.3) years who had never had periodontal therapy before. The results showed the presence of a severe form of this disease, stage III and IV, in 85.4% and 48.5%, respectively. Stages were assessed according to the 2017 classification of the American Academy of Periodontology and the European Federation of Periodontology (AAP/EFP). The authors presented the following risk factors for the occurrence of untreated periodontal disease: age ≥ 35 years, male sex, lower income, lower education, smoking, uncontrolled diabetes, and lower stress perception [3]. In another study by Relvas et al. (2022), data were collected from the clinical records of patients who attended the University Gandra Clinic in 2021 and 2022. Of 941 patients, 457 (48.6%) had periodontitis, 253 (26.9%) had gingivitis, and the remaining 231 (24.5%) had none of the listed diseases [4]. In their study, Janakiram et al. (2020) noted another important aspect of periodontitis by describing the disease from the public health perspective, reporting that at least 10% of adults worldwide may suffer from severe periodontitis. Additionally, the authors pointed out the global nature of the disease and, at the same time, the low public awareness [5].

One of the important factors for the course of the periodontitis is diabetes which is one of the most common social diseases. Diabetes affects about 60 million people in Europe, including approximately 10.3% of men and 9.6% of women aged 25 years and over [6]. The International Diabetes Federation predicted that by 2045, the number of people with diabetes will increase to 700 million worldwide. Furthermore, nearly half of adults with type 2 diabetes are unaware that they have the condition [7]. In addition to the negative effects on individuals, diabetes is a great burden for the economy and health care system. It is a chronic, long-lasting disease with consequences such as cardiovascular disease, kidney dysfunction, loss of vision, limb amputation, and death [6]. 

From an economic point of view, the treatment of complications of diabetes and periodontal disease is a major challenge for the entire health care system. The introduction of appropriate preventive and therapeutic approaches for both disease entities can have a measurable impact and significantly reduce treatment costs. Due to the proven bidirectional relationship between diabetes and periodontal disease, the authors would like to point out that patients suffering from periodontal disease and diabetes at the same time should constitute a special group of patients requiring comprehensive care with cooperation between diabetologists and periodontists. In the case of diabetic patients with periodontal disease, it seems justified to search for additional therapeutic methods. If these, at the same time, can have a beneficial effect on diabetes and periodontal disease, then the profit becomes multiplied. 

Hyperbaric oxygen therapy (HBO) is a method used to treat diabetic complications such as impaired wound healing and ulceration [8]. There are also numerous studies that describe the exact mechanism of the effect of hyperbaric oxygen on the condition of diabetic patients. A review by Resanović et al. (2020) explains the mechanism of HBO in the treatment of vascular complications in patients with type 2 diabetes. Hyperbaric oxygen therapy increases the solubility of oxygen in plasma, contributing to better oxygen diffusion to distant sites and preserving tissue viability (reversibly damaged by atherosclerosis-induced ischemia) along with microcirculatory restoration. Consequently, HBO exerts anti-atherosclerotic, antioxidant, and cardioprotective effects. Furthermore, it interacts with molecules involved in the regulation of nitric oxide synthesis and, thus, exerts anti-inflammatory and angiogenic effects in diabetic patients [9]. The aim of this study was to evaluate the efficiency of additional hyperbaric oxygen therapy used for periodontal disease treatment in patients with type 2 diabetes, in comparison to standard mechanotherapy.

## 2. Materials and Methods

### 2.1. Sampling and Sample Size

The study was conducted with patients with type 2 diabetes and periodontitis living in the Silesian voivodeship. Approvals were obtained from the Bioethics Committee at the Silesian Medical Association in Katowice, Poland (no. 39/2019) and the Complex of Municipal Hospitals with the Hyperbaric Oxygen Centre in Chorzów, Poland (no. KNW-0724/39/2019). Fourteen patients qualified for the study, 5 women and 9 men aged 57–89 years (mean age 69.6 ± 8.0), and were divided into a control group and a study group of 7 patients each. Glycated haemoglobin (HbA1c) levels ranged from 5.58% to 11.6%. 

A total of 369 periodontal pockets in the study group and 431 in the control group were measured. After measurements in the study group, 318 pockets were classified as moderately deep (4–5 mm) and 51 as deep (≥6 mm). In the control group, 357 periodontal pockets were classified as moderately deep (4–5 mm) and 74 as deep (≥6 mm).

### 2.2. Participant Recruitment

All patients were diagnosed with periodontitis, based on clinical assessment of the level of clinical attachment level (CAL) on examination of the tooth perimeter, with a reference point at the level of the cemento–enamel junction (CEJ). The study assessed disease stage and progression according to the 2017 AAP classification, which was developed at the World Workshop on Classification of the American Academy of Periodontology (AAP) in collaboration with the European Federation of Periodontology (EFP). A significant change introduced to the classification was the transfer of a system analogous to that used in oncology—TNM (tumour, nodus, metastases)—to periodontal conditions. Similar to the TNM classification, which is used to describe the progression and then decide on a therapeutic method, the periodontal disease classification was divided into stages, which defined the severity, complexity, and extent of the condition at the time of patient presentation, and grades, which provided additional information on the likely rate of disease progression and response to the introduced treatment [10].

The presence of a maximum interproximal CAL was used to assess stage. Factors influencing the need for complex rehabilitation due to teeth drifting, bite collapse, and the presence of fewer than 10 opposing pairs were then considered. A risk factor based on the HbA1c level was taken into account in the grade of periodontal disease assessment. To measure and document the periodontal status, an electronic study card was used, which was created by Christoph A. Ramseier (Department of Periodontology at the University of Bern, Switzerland) and translated into 32 languages [11]. The study was conducted by one dedicated person only, using an ISO 21672—compliant UNC15 probe (Hu—Friedy Mfg. Co., LLC., Frankfurt am Mein, Germany). Probing depth (PD), clinical attachment level (CAL), and the extent of inflammation using the bleeding on probing index (BOP) according to Ainamo and Bay (1975) were assessed. 

Patients were informed in detail about the purpose of the study and subsequently signed written consent forms for the study. A diagnosis with the usage of orthopantomography photos was carried out for each person. Patients who had undergone radiotherapy or chemotherapy in the past 12 months were excluded, as were patients who were immunosuppressed or malnourished, with avitaminosis, infectious diseases, blood diseases, osteoporosis, or allergies, alcohol abusers, nicotine users, pregnant women, and patients treated for periodontal disease in the past six months. 

According to a clinical examination by a hyperbaric physician, contraindications to HBO were used as additional exclusion criteria. The only absolute contraindication to hyperbaric oxygen therapy was untreated pneumothorax. The others belong to the relative contraindications, where the following can be distinguished: inability to compensate for middle ear pressure, fever, claustrophobia, pregnancy, severe heart failure, uncontrolled asthma, and concurrent chemotherapy, which may increase oxygen toxicity. The above criteria in place at the listed Hyperbaric Oxygen Centre are described by the European Wound Management Association (EWMA) in the paper “Use of Oxygen Therapies in Wound Healing” (2017) and are available on the European Underwater and Baromedical Society (EUBS) website [12].

Patients had to meet the following requirements: the presence of palpable CAL on periodontal examination on the proximal surfaces of at least two non-adjacent teeth, a diagnosis of type 2 diabetes, and a dentition of at least 10 teeth.

### 2.3. Study Design

The study group received standard mechanotherapy consisting of scaling with root planing (SRP—scaling and root planing) with irrigation with 0.2% chlorhexidine solution—CHX (Curasept S.p.A., Saronno, Italy) and received 30 sessions of HBO hyperbaric oxygen therapy (one patient had 27 sessions due to medical conditions). The sessions took place in a multi-person chamber, where the pressure was 2.5 ATA (253.31 kPa) and patients breathed 100% oxygen for 60 min. The exact course of the session was as follows: 10 min compression, 20 min oxygen breathing, 5 min break, 20 min oxygen breathing, 5 min break, 20 min oxygen breathing, and 10 min decompression. The regimen was based on the protocol in place for diabetic patients at the Hyperbaric Oxygen Centre where the study was conducted. Dental procedures were performed under antibiotic prophylaxis in a single-dose regimen 2 g Amoxicillin (Duomox®, Cheplapharm, Greifswald, Germany) 60 min before the procedure in accordance with the recommendations (2018) of the Working Group of the Polish Dental Society and the National Antibiotic Preservation Program on the use of antibiotics in dentistry [13]. None of the patients reported a general medical history of allergy to penicillin.

In the control group, SRP with 0.2% CHX irrigation was used as the only therapeutic method. During the first visit, a detailed hygiene instruction and a periodontal examination after the removal of supragingival deposits were performed. Once the patient qualified, non-surgical treatment of SRP was carried out with an ultrasonic scaler (Piezon250 EMS) using standard tips. The criteria for thorough cleaning of the subgingival area were a smooth root surface free of bacterial plaque and calculus. A further examination was performed after 8 weeks, which is the time required for tissue healing after non-surgical treatment [14]. 

Due to the lack of studies or commonly used management protocol among patients with periodontal disease and diabetes, the authors defined their own management protocol based on similar studies conducted on patients with periodontal disease only.

### 2.4. Diagnosis and Assessment of Periodontal Disease

Periodontal disease was diagnosed by assessing the level of clinical attachment level. The following criteria were taken into account:presence of CAL on the proximal surfaces of ≥2 non-adjacent teeth;presence of CAL ≥ 3 mm and PD pocket > 3 mm on the vestibular or lingual surfaces of ≥2 teeth, and the observed CAL was not due to other phenomena, i.e., recessions of traumatic origin, misalignment of the last molar, vertical root fracture (VRF), and endodontic lesion.

A four-stage scale to assess the stages was used:stage I—initial periodontitis;stage II—moderate periodontitis;stage III—severe periodontitis with possible additional tooth loss;stage IV—severe periodontitis with extensive tooth loss with the possibility of further tooth loss.

The stages provided additional information to determine individual needs based on a three-stage progression rate scale:grade A—slow rate of progression;grade B—medium rate of progression;grade C—fast rate of progression [10].

Evaluation of bleeding on probing (BOP) involves inserting a periodontal probe into the periodontal pocket with a force of 0.25 N. The presence or absence of bleeding is checked 10–15 s after the withdrawal of the probe. Measurements are taken at six points at each tooth. The percentage of sites with bleeding present in relation to all sites probed is then calculated.

Probing depth (PD) assessment involves inserting a periodontal probe into the gingival crevice or periodontal pocket with a force of 0.25 N. The measurement is read as the greatest value between the gingival margin and the bottom of the pocket. The measurements are taken at 6 points at each tooth. Depths of up to 3 mm are indicative of a healthy periodontium.

Assessing the level of CAL consists of inserting a periodontal probe into the gingival crevice or periodontal pocket with a force of 0.25 N. In a healthy periodontium, during examination, the cemento–enamel junction (CEJ) is not perceptible in the form of a slight roughness or fault between the smooth enamel surface and the rougher root surface. In periodontitis, a change in the position of the connective tissue attachment is observed. When the CEJ is visible, CAL is calculated by summing the distance from the CEJ to the gingival margin and the probing depth of the PD. On the other hand, when the CEJ is palpable subgingivally, the distance between the CEJ and the gingival margin is subtracted from the PD probing depth.

### 2.5. Data Analysis

Before the final analysis, a check was performed to ensure that the data collected were eligible for further statistical treatment, as we intended to use Student’s *t*-test for independent samples when analysing the results. Two groups were used for the study and the variables studied were quantitative. Graphically, it was checked that the variables under study were comparable to a normal distribution. An example of a graph from which the normality of the waveform was inferred is shown below (Figure 1).

Using the Fisher-Snedecor test, the assumption of homogeneity of variance was tested. Results below the critical value were obtained for all variants (Table 1).

Using the chi-square test for a significance level of *p* = 0.05 and 1 degree of freedom, the homogeneity of the group sizes was tested. The result of the test was below the critical value (Table 2).

## 3. Results

The results of the study showed a greater mean PD reduction and CAL gain in the study group with HBO than in the control group for both periodontal pocket types (moderately deep and deep). After comparing the results from both groups by using Student’s *t*-test, a significant difference in the change in PD and CAL was noted for the moderately deep and deep pockets for a significance level of *p* < 0.05. For all cases, the result was higher than the critical value, so it is permissible to claim that the difference between the research and control groups was significant (Table 3). The largest difference statistically was obtained for moderately deep pockets (4–5 mm) for the CAL parameter:control group—average CAL gain = 0.56 mm (X1);study group—average CAL gain = 0.92 mm (X2).

A comparison of mean BOP before and after treatment was also performed. Improvement was noted in both groups (Table 4). However, due to the large discrepancy between the study and control groups, the authors recommend approaching the results with caution and performing additional research.

## 4. Discussion

The study presented in this article is the first approach to assess the impact of HBO on the periodontium in a group of patients with diabetes. Oral diseases affect nearly 3.5 billion people worldwide. They encompass a range of conditions that include tooth decay, gum and periodontal disease, and oral cancer [6]. 

Periodontitis is a chronic inflammatory disease of multifactorial aetiology associated with a dysbiotic biofilm. The most common symptom is progressive destruction of the tooth-supporting apparatus. In the process of periodontitis, molecular pathways are activated, which results an increased production of proteases by the host. This leads to homeostatic imbalance and the destruction of periodontal ligament fibres, apical migration of the connective epithelium, and apical spread of biofilm along the root surface. In clinical practice, this manifests itself as a change in the level of position of the CAL, radiographically assessed alveolar bone loss, the presence of periodontal pockets, and gingival bleeding [15].

The risk factors of periodontal disease can be divided into two groups: modifiable and unmodifiable factors. The modifiable group includes socioeconomic status, non-specific and specific bacterial infections (non-specific bacterial biofilm, specific periopathogens and dysbiotic bacteria), smoking, lifestyle, inadequate diet with low calcium and vitamin D content, and certain general diseases such as diabetes, obesity, metabolic syndrome, and osteoporosis. Among the unmodifiable factors are age, gender, and genetic conditions (antimicrobial immune-inflammatory response, genetic–environmental interactions) [16].

### 4.1. Impact of Diabetes on Periodontal Disease

The first reports on the adverse effects of diabetes on the condition of periodontal tissues appeared in 1991. At that time, Loe named periodontal disease as the sixth complication of diabetes. Periodontitis was mentioned among conditions such as retinopathy, nephropathy, and micro and macroangiopathy, with their complications such as cardiovascular disease, neuropathy, and impaired wound healing [17]. Data from the literature show that patients with diabetes and a severe form of periodontitis have a 3.2 times higher risk of death in comparison to those with no periodontitis or a mild form [18]. A meta-analysis by Nascimento et al. (2018) found that diabetes increases the risk of periodontitis by up to 86% [19].

The relationship between diabetes and periodontitis is bidirectional; therefore, prophylaxis conducted by diabetologists and dentists in a multi-directional way is desirable. In diabetes, a longer duration of inflammation is observed. The consequence is longer osteoclast activity and bone resorption, resulting in a longer period of periodontal destruction [20]. Other potential influences from diabetes on periodontal disease are:increased production of free oxygen radicals;gingival epithelial barrier modifications;shifts in the dental plaque microbiome.

Moreover, diabetes leads to the formation of advanced glycation end-products (AGE) in periodontal tissues and their interaction with receptors (RAGE), which may be the root cause of inflammatory response activation and periodontal tissue damage [21]. A case–control study by Mesia et al. (2016) compared the immune-inflammatory response in periodontitis in patients with type 2 diabetes and a control group of patients without diagnosed diabetes. In the results, people with diabetes showed statistically significantly higher unstimulated levels of interleukin 6 (IL-6), IL-1β, tumour necrosis factor α, (TNF-α) interferon γ, IL-10, IL-8, macrophage inflammatory protein 1α (MIP1α), and 1β (MIP1β), and higher stimulated levels of IL-6, IL-8, IL-10, MIP1α, and MIP1β compared to the control group [22].

### 4.2. Impact of Periodontal Disease on Diabetes

A postulated mechanism linking periodontitis and diabetes is the influence of periopathogens and inflammatory mediators released locally in periodontal tissues. Then, as a result of their presence in the bloodstream, a systemic inflammatory response is triggered. In consequence insulin signalling and insulin resistance are impaired. Additionally, increased levels of HbA1c contribute to a higher risk of diabetic complications [23]. In a randomized controlled trial, a reduction in HbA1c levels after periodontal treatment was observed—a reduction of approximately 3–4 mmol/mol (about 0.3–0.4%) over a period 3–4 months after periodontal treatment. It is likely that the reduced plaque may be responsible for the reduction in the inflammatory process in the periodontal tissues, which reduces systemic inflammation and the activation of the immune system [24]. 

### 4.3. Hyperbaric Oxygen Therapy-Medical Applications

The action of hyperbaric oxygen is to supply oxygen at a high pressure to tissues that are hypoxic. This condition is observed in disease states with impaired oxygen supply to the tissues. Hypoxia occurs as a result of impaired oxygen transport to the cells, for example, as a consequence of vascular damage, circulatory insufficiency, circulatory trauma, or infection. In most cases, HBO is used as an adjunctive therapy. The mechanism of action allows the flushing out of toxic gases, stimulates fibroblast proliferation and neovascularisation, increases bacterial killing by a free radical mechanism, and suppresses inflammation by inhibiting leukocyte adhesion to the endothelium. Increased blood oxygen content allows recovery from hypoxia and circulatory disorders, enhances wound healing and the regeneration of damaged tissues, and improves fibrous scar remodelling. The aforementioned mechanisms help in the treatment of refractory conditions such as osteoarthritis, radiation necrosis, or diabetic foot. Increasing the amount of dissolved oxygen in tissue with a deficit of it creates a favourable environment for wound reconstruction and healing [8].

### 4.4. Hyperbaric Oxygen Therapy-Study Rewiev

In dentistry, HBO is used in areas such as osteoradionecrosis, adjunctive therapy for implant procedures in irradiated regions, osteomyelitis, and periodontal disease [25]. The impact of HBO in the treatment of periodontal disease as an adjunctive method was studied by several authors. Chen et al. (2002) considered the probing depth (PD), attachment loss (AL), plaque index (PLI, Silness and Loe 1964), and gingival indices (GI, Loe 1967). Measurements were taken at the first visit, immediately after the last HBO session, and one year after the end of HBO. The results showed statistically significant differences in GI, SBI, PD, AL, and PLI in the study groups compared to the control group. The authors additionally performed gingival blood flow measurement (GBF) and bacteriological examination. Later, Chen et al. (2012) confirmed the earlier results with a two-year follow-up period. The authors obtained a mean reduction of 0.23 mm in PD and gain of 0.2 mm in CAL in the study group in comparison to the control group. Wandawa et al. (2017) conducted a study with patients with chronic periodontitis assessing PD, CAL, and BOP. Patients were divided into groups treated with SRP alone or additionally with HBO sessions (8 or 16 sessions). A follow-up examination was performed after 15 and 30 days. The eight-session treatment group achieved a PD reduction and CAL gain of 0.82 mm and 1.13 mm, respectively, in comparison to the control group (treated with SRP only). On the other hand, patients treated with the 16-session regimen showed a change in the aforementioned parameters of a 1.13 mm reduction and 1.23 mm gain, respectively, compared to the control group. The authors concluded that the therapy was an effective adjunctive method. A subsequent study by Lombardo et al. (2020) assessed PD, CAL, BOP, and VPI (visible plaque index) before treatment, after two weeks, after six weeks, and after three months. The calculations showed no statistically significant differences between the groups for the PPD and CAL parameters. Only a statistically greater reduction was obtained for (*p* < 0.05) BOP in the study group compared to the control group. The authors concluded that HBO therapy in combination with subgingival mechanotherapy can be an effective treatment for moderate to severe forms of periodontitis [26]. A study by Mulawarmanti et al. (2019) in animals with diabetes and periodontitis showed a statistically significant reduction in IL-1β expression and an increase in IL-10 expression after treatment with HBO and HBO in combination with 3% Stichopus hermanii gel [27]. Giacon et al. (2021) presented a case report on the use of HBO in combination with advanced platelet-rich fibrin (A-PRF) as preparation for immediate implantation in a patient with advanced peri-implantitis [28].

The results presented by other authors positively assess the effect of hyperbaric oxygen therapy on the condition of periodontal tissues as a complementary method, which is in line with the conclusions of the present study. Currently, it is not possible to accurately compare work using HBO carried out on patients with periodontitis alone due to heterogeneity in the studies’ methodologies. However, the promising results of studies evaluating the use of HBO in diabetes justify the advisability of the above work. 

According to the authors of this article, HBO could be an effective and promising adjunctive treatment for periodontitis in a group of patients requiring additional care, namely those with type 2 diabetes. However, a cautious approach to the results and consideration of potential methodological flaws are recommended. The small group size was because most of the patients with type 2 diabetes did not meet the criteria of 10 teeth in the mouth. This suggests that the oral condition of diabetic patients is much worse than was assumed when creating the study protocol. Further studies conducted on a larger number of patients with a sufficiently long follow-up period are needed. The authors are aware that the present pilot study has some limitations, represented by the short-term assessment (2 months of follow-up) and small sample size. A single-centre setting involving a university dental clinic may have introduced an important bias due to which our results cannot be generalized. 

## 5. Conclusions

Data from the literature indicate that periodontal disease is more common than cardiovascular disease [5]. Furthermore, in addition to pain and impaired masticatory function and speech, it causes negative socio-economic consequences. The prevention and diagnosis of periodontal disease depend on the improvement of available preventive and therapeutic methods. The public health system lacks sustainable and effective procedural solutions operating in real-world settings. To meet the emerging needs of modern medicine, the European Federation of Periodontology (EFP), together with the International Diabetes Federation (IDF), have created practice guidelines for healthcare professionals in the management of patients with diabetes or periodontal disease. They list the actions of medical practitioners, which should consist of:taking a history of periodontal disease;making patients aware of the relationship between the two diseases;referring newly diagnosed patients for periodontal evaluation;cooperating with the dentist;motivating patients to make regular dental visits.

Dental professionals should:inform patients about the relationship between the two conditions;carry out regular periodontal check-ups;ask about HbA1c levels;cooperate with the general practitioner (GP), internist, or diabetologist and consider assessing the risk of diabetes in patients with suspected diabetes (for example, using a screening questionnaire) [29].

In the present study, the introduction of hyperbaric oxygen treatment was shown to have the potential to be an effective adjunctive treatment option, especially in patients with diabetes and advanced periodontitis. Due to the small sample size and the scale of the study limited to one centre, the authors suggest continuing the study. Currently, subgingival mechanotherapy is still the gold standard for the non-surgical treatment of periodontal disease.

## Figures and Tables

**Figure 1 healthcare-11-01344-f001:**
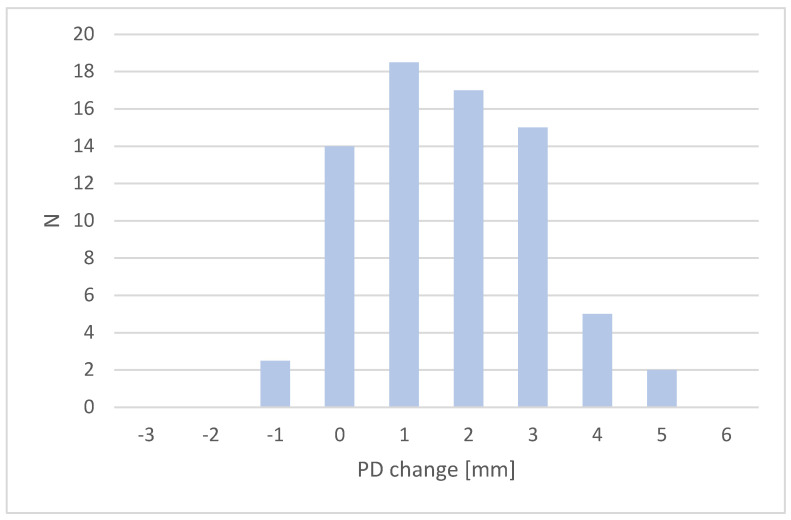
Distribution of the PD change for deep periodontal pockets (≥6 mm) in control group.

**Table 1 healthcare-11-01344-t001:** Fisher—Snedecor test results for variance homogeneity.

Parameter	Pockets	SD1	SD2	F-S
PD	4—5 mm	0.93	0.83	1.11
	≥6 mm	1.37	1.43	1.04
CAL	4—5 mm	1.01	1.07	1.06
	≥6 mm	1.34	1.68	1.25

SD1—standard deviation of the parameter for the control group; SD2—standard deviation of the parameter for the study group; F-S—Fisher-Snedecor test results.

**Table 2 healthcare-11-01344-t002:** Chi-square test results for the homogeneity of the group sizes.

Group	Pockets	N	E	Chi-Square
Control Group	4–5 mm	357	337.5	0.48
	≥6 mm	74	62.5	0.17
Study Group	4–5 mm	318	337.5	0.48
	≥6 mm	51	62.5	0.17
			Sum	1.28

N—group size; E—expected value.

**Table 3 healthcare-11-01344-t003:** T-Student test results for control group and study group comparison.

Parameter	Pockets	X1 (SD1)	X2 (SD2)	T-Student
PD	4–5 mm	0.79 (0.93)	0.99 (0.83)	2.88
	≥6 mm	1.70 (1.37)	2.79 (1.43)	4.27
CAL	4–5 mm	0.56 (1.01)	0.92 (1.07)	4.41
	≥6 mm	1.09 (1.34)	1.87 (1.68)	2.90

**Table 4 healthcare-11-01344-t004:** Mean BOP comparison between control group and study group.

Group	BOP1	BOP2
Control Group	75.4%	63.7%
Study Group	32.0%	28.8%

BOP1—mean BOP measured during the first visit; BOP2—mean BOP measured during the control visit (after 8 weeks).

## Data Availability

Data will be available upon request from the corresponding author.

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
