# Peer review of "Pilot Study: The Effectiveness of Hyperbaric Oxygen Therapy in the Treatment of Periodontitis in Patients with Type 2 Diabetes"

_healthcare, 2023, doi:10.3390/healthcare11091344_

Round 1

Reviewer 1 Report

very good job. however the title could be more explicit by citiyng periodontitis

Reviewer 2 Report

The authors should be congratulated on this manuscript and insightful clinical study.  The study is promisng as a pliot trial and this reviewer suggests that the authors persue a larger =, multi-center study to validate these results. The results are very interesting and suggest another area of benefit to patients ravaged by diabetes.   

2 questions/comments:

1.  Why was a non-standard HBOT treatment protocol used (2.5 ATA for 60min).  I would suggest consideration of optimizing the protocol and using a standard table of 2.4-2.5 ATA for 90min, 30min O2 with airbreaks (30-5-30-5-30). This would increase the oxygen exposure and potentially improve the results.  The rational for using the non-standard table should be explained in the manuscript.

2.  What type of DM management protocol was used during the study.  It would be interesting to see if there was any change in the HgA1C at the end of the trial.

Reviewer 3 Report

Line 94

Reported: Glycated haemoglobin (HbA1c) levels ranged from 94 5.58% to 11.6%.

Comments: The American Diabetes Association (ADA) 2023 recommendation, while reaffirming the usefulness of glycated hemoglobin (Hb1aC) assessment - Recommendation 2.1b, points out that the prognosis and the choice of diabetes therapy mainly depend on the daily variability of glycemia (the oscillation between high and low values must be minimal) and the cardiovascular (for example, heart failure) and renal (monitor eGFR for renal insufficiency) complications - ADA Recommendation 9.4b, Section 10 and Section 11.

Proposed change: Specify the distribution of cardiovascular and renal complications, the frequency of hypoglycemic crises in the observed patient population.

References:

Nuha A. ElSayed and others on behalf of the American Diabetes Association, Summary of Revisions: Standards of Care in Diabetes—2023. Diabetes Care 1 January 2023; 46 (Supplement_1): S5–S9. https://doi.org/10.2337/dc23-Srev

 Line 105

Reported: A risk factor based on HbA1c level was taken into account in the grade assessment.

Proposed change: see previous comment (Line 94)

Line 119

Reported: According to a clinical examination by a hyperbaric physician, contraindications to HBO were used as additional exclusion criteria.

Proposed change: Indicate the guideline used as a reference for the assessment of contraindications to HBOT (e.g. EUBS, UHMS, SPUMS or other).

 Line 129

Reported; The sessions took place in a multi-person chamber, where the pressure was 2.5 ATA and patients breathed 100% oxygen for 60 minutes.

Proposed change: Add the internationally agreed unit of pressure in parentheses: 2.5 ATA (253.31 kPa)

Line 253

Reported. A systematic review by Baitule et al. (2021) summarizes the effect of HBO on glycemic levels in patients with diabetes. It comes up with a conclusion that HBO causes lower blood glucose levels and improved insulin sensitivity in type two diabetes. However, the authors point out the need for further studies on a large group of patients. They positively summarize its effect as an adjunctive method in selected cases [22]. In a review by Sharma et al. (2021) showed that HBO was effective in completely healing diabetic foot ulcers and reducing the rate of major amputations. According to the authors, hyperbaric oxygen therapy is effective as an adjunctive treatment for diabetic foot ulcers. They point out as well to consider the methodological flaws of all studies and the need for further multidisciplinary studies, with sufficiently large sample sizes and appropriate methodology, to evaluate the efficiency and safety of HBO as a complementary treatment [23].

Comments. The literature on HBOT in diabetes, in a general sense and not directly for periodontal disease, is vast and controversial. It needs to be analyzed entirely and not only partially.

Proposed change. Delete this part of the literature review on the effects of HBOT on diabetes, in a general approach.

Line 313 References

Comments. Four manuscripts out of 23 reported in the references are dated more than 10 years ago.

Proposed change. If not essential to support the content, delete or replace them with manuscripts published by the same Authors more recently.
